# Bayesian nonparametric discovery of isoforms and individual specific quantification

Derek Aguiar[1], Li-Fang Cheng[2], Bianca Dumitrascu[3], Fantine Mordelet[4], Athma A. Pai[5,7] &
Barbara E. Engelhardt [1,6]

Most human protein-coding genes can be transcribed into multiple distinct mRNA isoforms. These alternative splicing patterns encourage molecular diversity, and dysregulation of isoform expression plays an important role in disease etiology. However, isoforms are difficult to characterize from short-read RNA-seq data because they share identical subsequences and occur in different frequencies across tissues and samples. Here, we develop BIISQ, a Bayesian nonparametric model for isoform discovery and individual specific quantification from short-read RNA-seq data. BIISQ does not require isoform reference sequences but instead estimates an isoform catalog shared across samples. We use stochastic variational inference for efficient posterior estimates and demonstrate superior precision and recall for simulations compared to state-of-the-art isoform reconstruction methods. BIISQ shows the most gains for low abundance isoforms, with 36% more isoforms correctly inferred at low coverage versus a multi-sample method and 170% more versus single-sample methods. We estimate isoforms in the GEUVADIS RNA-seq data and validate inferred isoforms by associating genetic variants with isoform ratios.

[1] Department of Computer Science, Princeton University, Princeton, NJ 08540, USA. [2] Department of Electrical Engineering, Princeton University, Princeton, NJ 08540, USA. [3] Lewis-Sigler Institute, Princeton University, Princeton, NJ 08544, USA. [4] Institute for Genome Sciences and Policy, Duke University, Durham, NC 27708, USA. [5] Department of Biology, Massachusetts Institute of Technology, Cambridge, MA 02139, USA. [6] Center for Statistics and Machine Learning, Princeton University, Princeton, NJ 08540, USA. [7] Present address: RNA Therapeutics Institute, University of Massachusetts Medical School, Worcester, MA 01605, USA. Correspondence and requests for materials should be addressed to D.A. (email: daguiar@princeton.edu) or to B.E.E. (email: bee@princeton.edu)

Alternative splicing is the process by which a single gene produces distinct mRNA isoforms, which vary in usage of component exons[1]. Isoforms can differ by alternative transcription initiation sites, alternative usage of splice sites (either 5′ donor or 3′ acceptor sites), alternative polyadenylation sites, or variable inclusion of entire exons or introns (Fig. 1). Altogether, alternative splicing enables the large diversity of mRNA expression levels and proteome composition observed in eukaryotic cells, which is particularly important for regulating the context-specific needs of the cell[2].

It is estimated that 95% of human protein-coding genes can be alternatively spliced[1]. These splicing decisions are important drivers of many biological processes, with considerable variation in splicing patterns across human tissues[3]. For example, mutations in splicing regulatory elements may lead to disease pathogenesis and progression,[1, 4–8] and mutations in protein domains of specific splicing factors occur at a high rate in tumor cells, resulting in increased cellular proliferation[9]. Furthermore, proteins resulting from splicing variants often have distinct molecular functions. For instance, the two variants of survivin have opposite functions: one with pro-apoptotic and the other with anti-apoptotic properties[10].

Although there is increasing evidence of the biological importance of splicing processes, the precise role of alternative isoforms in regulating complex phenotypes is still largely uncharacterized. This gap in understanding is due, in part, to the difficulty of identifying and quantifying isoforms with high accuracy from short-read RNA-seq data[11]. Transcript reconstruction is essential to elucidate the role of gene expression in biological processes because gene-level quantification is convoluted by the multiple transcribed isoforms for each gene. The difficulties in isoform quantification stem from the tissue- and sample-specific composition and expression patterns of isoforms, the lack of a complete reference for isoform composition, and low abundance levels of many isoforms[2]. Further, RNA-seq reads that

overlap informative splice junctions are rare, often noisy[12], and difficult to map to a reference genome[13]. Improvements in reconstructing and quantifying tissue- and sample-specific isoforms would enable substantial improvements in understanding the role of alternative splicing in complex disease.

While many tools exist for isoform reconstruction using RNA-seq data, these methods have a number of drawbacks. First, many quantification methods assume that a high-resolution isoform sequence reference is available for each gene in the genome[14–16]; in practice these references are often not available or incomplete for non-model organisms and rare tissue or disease samples[11]. Second, while a few methods process multiple samples simultaneously[17–19], most methods consider a single sample in isolation, which fails to exploit the sharing of isoforms across samples to gain power for identification of rare or low abundance isoforms[20–22]. Third, many methods make technology-dependent assumptions by controlling for specific biases (e.g., non-uniform sampling of reads[23]) that do not generalize to mixtures of existing technologies or new technologies with different biases.

Our method, Bayesian isoform discovery and individual specific quantification (BIISQ), addresses these limitations. First, BIISQ uses annotations of transcribed regions as prior information[24, 25], but the number and composition of isoforms across samples are estimated directly from the data, and the catalog of isoforms may grow with additional observations. Second, BIISQ explicitly captures isoforms shared across samples using a Bayesian hierarchical admixture model, which models multiple samples jointly and borrows statistical strength across samples to identify shared isoforms that may be in low abundance. Third, BIISQ assumes that each nucleotide base in an isoform has an independent frequency in the mapped reads, allowing BIISQ to account for read mapping biases in RNA-seq data[26].

We develop a computationally tractable stochastic variational inference (SVI) algorithm to fit this model to short-read RNA-seq data to estimate the structure of isoforms, probabilistically assign reads to isoforms, and compute sample-specific and global isoform proportions[27]. We compare and validate BIISQ results on simulated data from the benchmarker for evaluating the effectiveness of RNA-seq (BEERS) software[16] and by simulating short-read data from available Pacific Biosciences (PacBio) Iso-seq data, which include 1–10 kb sequence reads potentially capturing full-length isoforms[28]. Finally, we apply BIISQ to a large RNA-seq data set from lymphoblastoid cell lines (LCLs)[29] to identify the catalog of isoforms across samples. We use our catalog of sample-specific inferred isoforms and genotype data to identify genetic variants associated with isoform ratios and then to assess the functional significance of alternatively spliced genes and associated splicing variants.

## Results

**Isoform reconstruction and quantification with BIISQ.** The goal of isoform reconstruction and quantification is to robustly estimate both absolute and relative mRNA isoform expression levels for transcripts expressed at both low and high levels, and for each sample in short-read RNA-seq data from multiple samples. Our method, BIISQ, approaches this problem by postulating a model of isoform composition and relative isoform abundance shared across samples. Specifically, BIISQ implements a Bayesian non-parametric hierarchical model of RNA-seq reads and isoforms inspired by the hierarchical Dirichlet process[30], and we use stochastic variational inference (SVI) methods for computationally tractable and robust posterior inference[27]. Importantly, in our model, exon usage, RNA-seq read assignments, and the sample-specific and global isoform proportions are interpretable

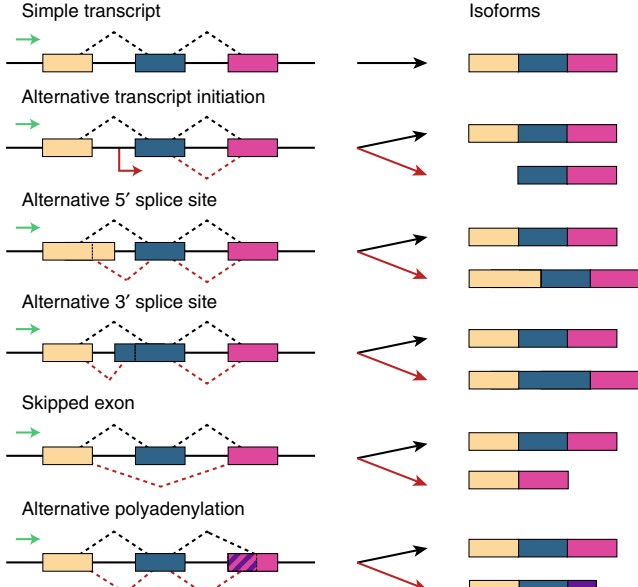

**Fig. 1** Alternative splicing mechanisms. A single gene may be transcribed into several distinct mRNA variants called isoforms through alternative splicing mechanisms. This figure shows six common types of splicing events (top to bottom): simple transcript; alternative transcription start site; alternative 5′ splice site; alternative 3′ splice site; skipped exon; and alternative polyadenylation

parameters that translate directly to isoform composition, and global and sample-level isoform quantification (see Methods).

The BIISQ model probabilistically maps each RNA-seq read into a distribution over isoform exon compositions. Sample-specific isoform proportions are drawn from a global distribution over an arbitrarily large catalog of isoforms. The exon composition of each isoform is modeled with a structured prior over exon usage that constrains the space of possible isoforms to those with support in the observed RNA-seq reads. The model grows the global isoform catalog by constructing novel isoforms given observed RNA-seq reads inconsistent with the current catalog.

Variational methods enable computationally tractable posterior inference in Bayesian models such as BIISQ[31, 32]. In brief, the posterior distribution of the BIISQ model is intractable to compute directly; instead, we hypothesize a family of tractable variational distributions. Then, we iteratively compute the values of the variational parameters that minimize the distance between the variational and true posterior distributions with respect to the Kullback–Leibler divergence[33, 34]. BIISQ implements stochastic variational inference (SVI), an extension of variational inference that uses random subsets of the samples to update the variational parameters[27].

**Related isoform quantification methods**. Methods for jointly inferring and quantifying alternatively spliced transcripts can be broadly partitioned based on the required level of reference annotation[35]. Transcriptome annotation-dependent methods require complete annotation of the transcriptome, including isoform transcripts and splice junctions[14–16]. In contrast, annotation-free methods require neither transcriptome nor genome annotations[20, 36, 37]. A third class of method requires annotations of transcribed regions but is agnostic to isoform and splicing annotations[20–22]; our method BIISQ is in this category. Methods may include modes that cross these categories. For example, Cufflinks has evaluation modes that can be annotation free or guided by reference annotations of transcribed regions[20].

We compared results from BIISQ with four representative isoform reconstruction and quantification methods: Cufflinks[20], CEM[22], SLIDE[21], and ISP[19]. These methods were selected based on the following criteria: (i) the ability to use annotations of transcribed genomic regions for isoform discovery and quantification, but no requirement for isoform transcripts or splice junction annotations; (ii) coverage of combinatorial and statistical approaches; (iii) support for both single-end and paired-end reads; and (iv) high-quality performance in a recent benchmark study of isoform detection and quantification[38]. While ISP does not leverage available gene annotations, it does support isoform reconstruction across multiple samples simultaneously.

Cufflinks uses a parsimonious approach to isoform discovery in order to find the minimal number of transcripts to explain the aligned reads[20]. After filtering erroneous spliced read alignments, aligned reads are assigned to vertices in an overlap graph, whose edges represent isoform compatibility between aligned reads. Transcript assembly then reduces to finding a minimum set of paths through the overlap graph such that each aligned read is part of a path. Transcript quantification uses a statistical model for RNA-seq reads to compute a point estimate of the isoform quantifications, extending an earlier unpaired model[39].

CEM, an extension of the method IsoLasso[40], constructs a connectivity graph to generate a set of candidate isoforms[22]. CEM and IsoLasso model the coverage of aligned reads at each location as a Poisson distribution and use lasso regression to produce a set

of inferred isoforms and abundance levels. The principle difference between CEM and IsoLasso is how candidate isoforms are selected: CEM uses expectation maximization (EM) while IsoLasso solves a quadratic program. In our comparison, we preferred CEM because of superior performance demonstrated on benchmark data[22].

The sparse linear modeling for isoform discovery and abundance estimation (SLIDE) method implements a statistical approach based on the start and end positions of aligned reads[21]. SLIDE computes the number of aligned read start and end positions that group into transcribed regions of exons and organizes them into bins. Isoform proportions are quantified using a linear model of the observed bin proportions; a modified lasso penalty limits the number and composition of isoforms. We ran SLIDE using two settings of the regularization parameter, $\lambda = 0.01$ (denoted SLIDE_more) and $\lambda = 0.2$ (denoted SLIDE_fewer), which encourages more and fewer discovered isoforms, respectively.

The iterative shortest path (ISP) isoform reconstruction method builds a multi-sample generalization of the connectivity graph often used for transcriptome assembly[19, 41]; vertices in this graph denote transcribed segments, and edges connect two segments if there exists one or more reads supporting their adjacency. In this model, every distinct path corresponds to a possible isoform segment. By assigning weights to edges that are inversely proportional to the probability of inclusion in an isoform, the problem of isoform reconstruction is equivalent to solving an iterative shortest path problem.

**Evaluation criteria**. We evaluated precision and recall for each method in terms of exact and partial matches to simulated RNA-seq data[42]. Precision and recall were calculated based on exact full-length isoform matches between simulated and estimated isoforms (Eq. (4), Methods). Partial precision and recall were calculated by defining imperfect matches between each estimated transcript and the true transcripts (see Methods and Supplementary Fig. 1). We controlled for issues regarding exon identification by counting an exon as successfully inferred if any subsequence of the inferred isoform overlapped an exon in the gene annotation. Thus, reconstructing any subsequence of an exon was equivalent to reconstructing the whole exon correctly. For matched transcripts, we also computed the proportion of isoform bases correctly covered (Supplementary Methods).

**Short-read RNA-seq simulations with BEERS**. We first evaluated our model on simulated data generated using the benchmarker for evaluating the effectiveness of RNA-seq software (BEERS)[16]. In our simulations, we varied the number of alternatively spliced transcripts, coverage, the number of samples, read lengths, and reference annotations (Methods and Supplementary Fig. 2). After removing genes with fewer than three exons, we divided the simulated genes into three equally sized groups according to exon counts, producing gene sets with 3–6 exons, 7–12 exons, and 13–182 exons. In total, the simulation produced reads for 3102 genes across 532,800 samples.

To test the accuracy of each method, we applied the five isoform reconstruction methods to these simulated data and computed the precision and recall of the isoform discovery results —both perfect and partial matches (Fig. 2). For both perfect and partial matches, BIISQ showed significantly higher precision across the 3102 genes (pairwise t-tests, all $p \leq 2.2 \times 10^{-16}$). However, ISP showed significantly higher recall at the cost of lower precision than both BIISQ and Cufflinks (pairwise t-tests, all $p \leq 2.2 \times 10^{-16}$).

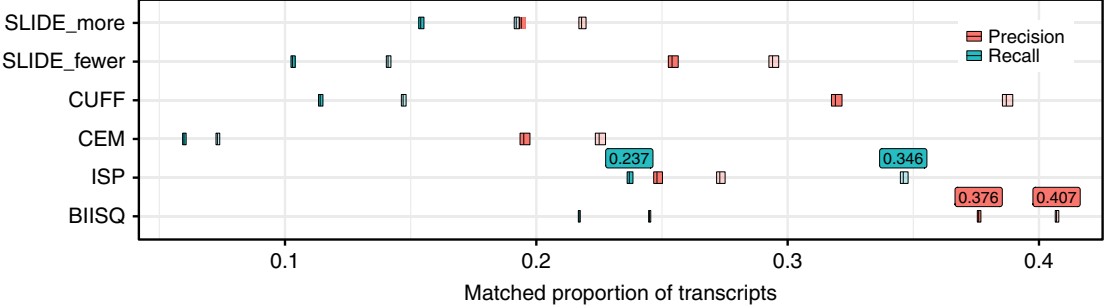

**Fig. 2** Isoform discovery precision and recall for simulated data. Precision (red) and recall (blue) of the results from BIISQ, ISP, CEM, Cufflinks (CUFF), and SLIDE (SLIDE_more and SLIDE_fewer) applied to the BEERS-simulated single-end RNA-seq data. The thick center bars denote the mean precision or recall and the fill denotes three times the standard error. Transparent fill denotes partial precision and recall with a matching threshold of 0.1. Across all methods, the best (partial) precision and recall values are annotated above their respective data points

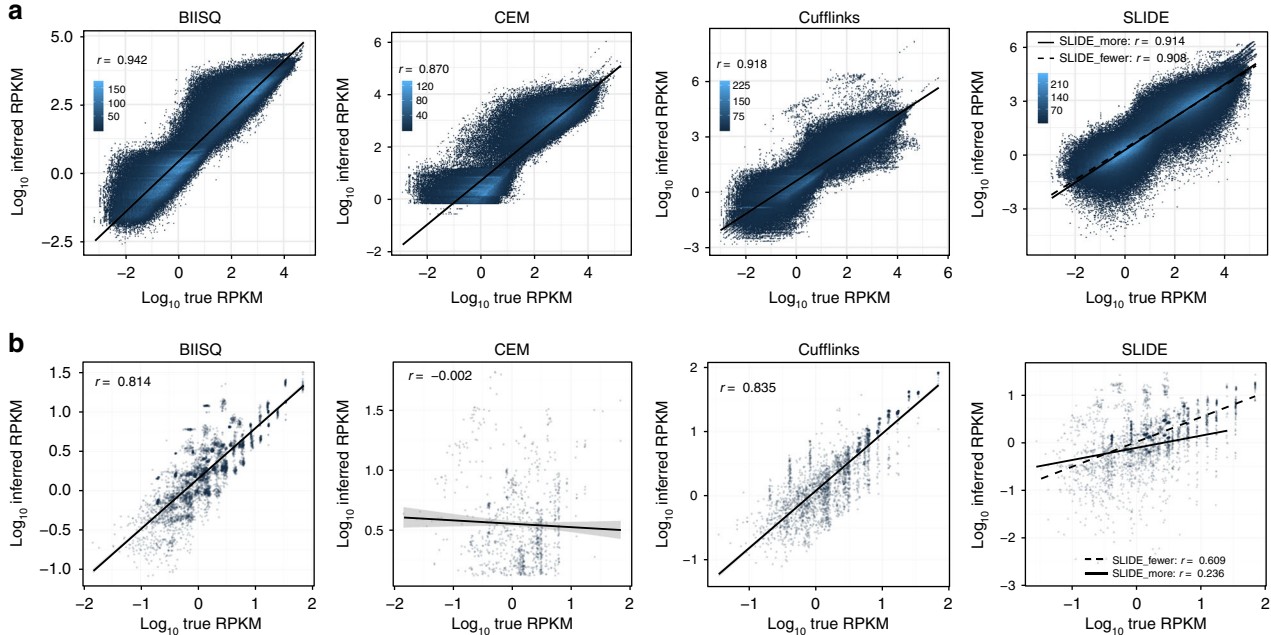

**Fig. 3** Isoform quantification accuracy. Correlation between true RPKM and inferred RPKM for **a** BEERS-simulated data and **b** Iso-Seq-simulated data. Spearman correlation coefficients for results from BIISQ, CEM, Cufflinks, SLIDE_more, and SLIDE_fewer were **a** 0.942, 0.870, 0.918, 0.914, and 0.908, respectively, for BEERS-simulated data and **b** 0.814, −0.002, 0.835, 0.609, and 0.236, respectively, for simulated short-read data from Iso-Seq reads. A regression line represents the best linear fit for each method to the expression data

BIISQ achieves the second highest recall for both perfect and partial matches.

We also evaluated isoform reconstruction based on terminal exons at both the 3′ and 5′ ends of the gene since isoform boundaries are more difficult to infer due to biological variation and known biases of RNA-seq[42]. When only the exons internal to the true transcription start and end sites are considered, the performance of all methods improves dramatically; BIISQ achieves the highest mean precision with 0.612, and ISP achieves the highest mean recall with 0.568, while the relative ranking of the methods in terms of precision and recall remains the same (Supplementary Fig. 3). When considering the proportion of exon sequence shared by matched inferred and true isoforms, BIISQ outperforms the other methods with 91.6% of true isoform exon sequence covered (Supplementary Table 1). This result is expected because BIISQ explicitly builds isoforms using a reference for the collection of exons within each gene.

The precision and recall results are largely recapitulated when we factored the results by the number of alternatively spliced transcripts (Supplementary Fig. 4) or the number of exons in the gene (Supplementary Figs. 5 and 6). All methods performed better for genes with a small number of alternative isoforms (1–4) and genes with a fewer number of exons (3–6), except for ISP, which had a higher recall for genes with 13–182 exons at threshold = 0.1. This is because ISP predicts a large number of isoforms, many of which are incorrect, as indicated by a consistently low precision (Supplementary Fig. 5). Moreover, BIISQ showed the largest difference between maximum and minimum precision (or recall) when factoring the results by the number of transcripts (Supplementary Fig. 4). This suggests that there are opportunities to improve performance for BIISQ in precision and recall for genes with large numbers of isoforms or exons. The BIISQ hyperparameters were set by grid search on a single gene with small numbers of alternative transcripts;

optimizing hyperparameters on a more diverse set of genes would likely improve precision and recall.

Next, to study BIISQ's high precision, we evaluated the number of perfectly inferred isoform transcripts with positive expression values across coverages, which is the number of bases sampled from the transcript with simulated reads normalized by the transcript length. Across all samples in the simulated data, BIISQ, CEM, Cufflinks, SLIDE_more, SLIDE_fewer, and ISP correctly inferred 246,896, 15,829, 90,186, 85,156, 59,716, and 242,244 transcripts, respectively, at a coverage of <1; the gains are more pronounced at a coverage of <0.1, where BIISQ inferred 11,153 more true transcripts than ISP and 26,465 more than Cufflinks (Supplementary Fig. 7). Despite the higher recall of ISP, BIISQ inferred more transcripts at lower expression levels, which highlights that much of the performance gains of BIISQ come from explicitly encoding gene composition and exon usage, and sharing strength across samples to identify low abundance isoform transcripts. These results show that BIISQ is more precise than the related methods, and BIISQ infers more total isoforms than all methods besides ISP, which has better recall than BIISQ or Cufflinks at the expense of greater numbers of false discoveries.

We assessed the quantification accuracy of each method by computing the correlation between true and inferred normalized read counts independently for each gene (reads per kilobase of exon per million mapped reads, or RPKM). We omitted ISP from these results as ISP reports expression as a percentage and relies on third-party methods to compute RPKM[19, 43]. BIISQ inferred positive expression for 697,605 transcripts compared to 556,306, 343,494, 316,975, and 179,327 for SLIDE_more, Cufflinks, SLIDE_fewer, and CEM, respectively. We found a wide range of expression-level estimates from the five methods (Fig. 3a), typical of isoform quantification in human samples[42]. Overall, BIISQ showed the highest correlation of expression across the BEERS data, followed by Cufflinks (Spearman correlation coefficient, BIISQ $r = 0.942$, Cufflinks $r = 0.918$). Correlation

improves for BIISQ, Cufflinks, and CEM for high coverage transcripts (Supplementary Fig. 8), suggesting that the difficulty associated with reconstruction of low coverage transcripts negatively affects quantification.

**Short-read RNA-seq simulations from long-read RNA-seq data.** The BEERS-simulated data models technology-specific biases of short-read RNA-seq data but does not capture the exon composition of true transcript isoforms. The Pacific Biosciences Iso-Seq protocol enables single molecule transcriptome sequencing with read lengths of up to 10 kb[28]. Iso-Seq reads may span entire RNA transcripts, making the characterization of isoform composition straightforward relative to inference from short-read data, where a short read may map to multiple isoforms. While long-read sequencing allows experimentally driven evaluation of isoform reconstruction, the cost and platform-specific error rates make this technology unlikely to replace short-read RNA-seq in the near future, necessitating the development of methods such as BIISQ. Further, while long reads facilitate isoform reconstruction, quantifying isoforms is challenging due to low throughput, making precision and recall the principal metrics for evaluation of Iso-Seq data[44].

We simulated short-read RNA-seq data from full-length Iso-Seq reads, which allows us to precisely capture true isoform composition and proportions in simulated data. To do this, we constructed a reference set of genes from the Iso-Seq human transcriptome reference samples of heart and brain tissue. After mapping genes and transcripts across tissues, we identified seven genes with two or more isoforms in the heart and brain tissues (see Supplementary Methods). To account for the platform-specific biases in Iso-Seq, we evaluated both GMAP and STARlong's Iso-Seq read alignments to the human genome version hg19[45] (Supplementary Figs. 9 and 10). For seven transcripts (Supplementary Table 2), we simulated reads with lengths 50, 100, and 200 bp for 50 replicates of brain and heart

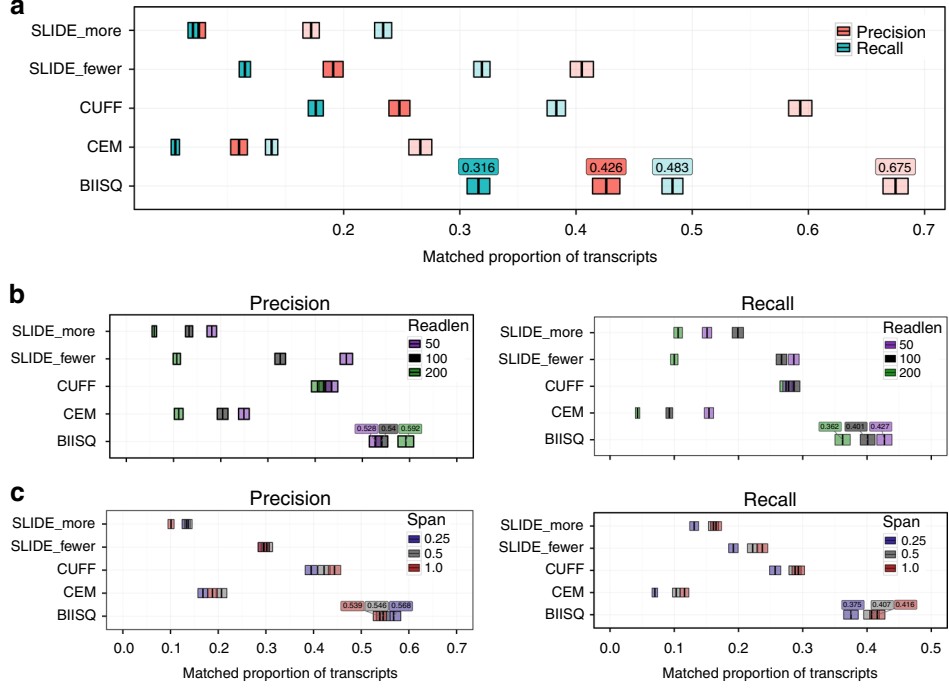

**Fig. 4** Comparison of methods on Iso-seq simulations. Precision (red) and recall (blue) of the results from BIISQ, CEM, Cufflinks (CUFF), SLIDE_more, and SLIDE_fewer applied to **a** the short-read data simulated from Iso-Seq reads; **b** simulated data split by read length; and **c** simulated data split by span. Transparent fill denotes partial precision and recall with a matching threshold of 0.2. The thick center bars denote the mean precision or recall, and the fill denotes twice the standard error. The best (partial) precision and recall values are annotated above their respective points

Iso-Seq samples. We define transcript span relative to the transcript sequence coverage of simulated reads; for example, a span of 0.5 indicates that short reads were generated until the reads mapped to half of the length of the Iso-Seq reads for a transcript.

We first evaluated the accuracy of isoform reconstruction in terms of perfect and partial (threshold = 0.2) precision and recall. Due to the complex error profile and high error rates in PacBio sequencing[46], we evaluated the four methods that support PacBio reference sequences: BIISQ, CEM, Cufflinks, and SLIDE. BIISQ achieved the highest precision and recall from both exact and partial matching thresholds across the seven Iso-Seq transcripts (Fig. 4a). This strong performance improvement remains when partitioning the RNA-seq data by read length and span (Fig. 4b,c). Importantly, the performance of BIISQ, Cufflinks, and SLIDE_fewer does not deteriorate substantially in either precision or recall for paired-end short reads relative to the deterioration in performance from CEM and SLIDE_more (Figs. 2 and 4a).

We then compared isoform quantifications across the four methods in the paired-end short-read data (Fig. 3b). BIISQ inferred 5395 transcripts while Cufflinks, SLIDE_fewer, SLIDE_more, and CEM inferred 3111, 2074, 1221, and 1003 transcripts, respectively. Rankings of quantification results on paired-end data largely mirrored the BEERS simulations, with Cufflinks and BIISQ achieving the highest correlation between true and inferred expression (Spearman correlation coefficients, Cufflinks $r = 0.835$ and BIISQ $r = 0.814$). BIISQ and Cufflinks showed the greatest agreement between any pair of methods (Spearman correlation coefficient, $r = 0.835$, Supplementary Fig. 11). BIISQ inferred more transcripts across all exon compositions (Supplementary Fig. 12) and all spans (Supplementary Fig. 13) in the Iso-Seq data.

We investigated the run time of each method as a function of the number of exons, gene length, read length, and span; BIISQ run times are averaged across 20 runs and eliminate the onetime cost to convert aligned reads to read terms (Supplementary Figs. 14–17). CEM was the most efficient method tested, followed closely by Cufflinks, while BIISQ and SLIDE had the longest run times. However, isoform reconstruction can be parallelized at the level of reference transcripts, so difficulties associated with running BIISQ transcriptome-wide may be reduced by using many compute nodes to process distinct genes in parallel.

**GEUVADIS RNA-seq data for 462 samples**. Building on the simulated data results, we tested BIISQ transcript reconstruction and quantification on high-dimensional RNA-seq data across ethnically diverse samples. We applied BIISQ to short-read RNA-seq data for 462 lymphoblastoid cell lines (LCLs) from the GEUVADIS RNA sequencing project for 1000 Genomes Project samples[29]. We built a model of transcription for each gene from the human genome annotations in GENCODE release 19 and mapped RNA-seq reads with STAR 2-pass to the human genome version hg19 (Methods). Applying BIISQ to these data, we discovered 31,712 novel and 14,044 known transcript isoforms with respect to the GENCODE database v19 transcript isoform annotations, considering only perfect matches to isoform exon composition. When using a matching threshold of 0.2, we discovered 24,871 novel and 20,885 known isoforms. The distribution of the number of isoforms per gene is peaked for genes with no evidence of alternative splicing (one transcript) and heavily spliced genes (≥7 transcripts), although this distribution could be confounded by erroneous splice junctions and fragmented transcripts in the BIISQ output (Methods and Supplementary Fig. 18)[12].

To investigate population- and sex-specific splicing patterns, we analyzed transcript ratio patterns across all genes in the GEUVADIS data. We considered global signatures of differential transcript ratio usage, and we did not find a significant difference in the average isoform transcript counts across sex ($\chi^2$ test, $p \leq 0.99$) or population ($\chi^2$ test, $p \leq 1$) when counts were aggregated across protein-coding genes. We computed population- and sex-specific transcript ratio distributions for each protein-coding gene separately using likelihood ratio (LR) tests (Methods). We found 924 and 148 genes that showed population- and sex-specific transcript ratio distributions, respectively ($\chi^2$ test, Bonferroni-corrected $p \leq 0.05$; Supplementary Data 1). The gene *PTPRN2* showed the most significant differential effects of population on isoform ratios (LR test, Bonferroni-corrected $p \leq 2.2 \times 10^{-16}$; Fig. 5a, top). The gene *LGALS9B* showed the most significant differential effects of sex on isoform ratios (LR test, Bonferroni-corrected $p \leq 2.2 \times 10^{-16}$; Fig. 5a, bottom, b). Most samples express at most two of the five isoform transcripts of *LGALS9B* (ordered by GENCODE annotation), but females show more variable isoform expression levels than males, in particular for isoform 5 (Fig. 5b).

Next, due to scarce information on population- or sex-specific transcript ratios, we validated these results by testing for over-representation of population- and sex-specific variants in the exonic and intronic regions of the 924 and 148 genes (Supplementary Data 1). First, we partitioned the GEUVADIS sample into subgroups by (a) 1000 Genomes Project population samples in GEUVADIS (CEU, TSI, FIN, GBR, YRI), (b) European (EUR) versus African (AFR) ancestry, and (c) sex (male, female). A variant allele is included if its minor allele frequency (MAF) in the GEUVADIS data is above a threshold, and if its MAF for individuals within the subgroup in question is above the population- or sex-specific threshold (Methods). We compared intragenic variant frequencies in the population- and sex-specific genes to variant frequencies in background genes randomly selected from genes that do not overlap the population- and sex-specific genes (version hg19) using the hypergeometric test for overabundance. To control for linkage disequilibrium (LD) among variants, we removed variants that were well correlated with neighboring variants (Methods). We found a significant over-representation of population-specific alleles at MAF ≥0.15 and a population threshold ≥0.15 (hypergeometric test, Bonferroni-corrected $p \leq 1.8 \times 10^{-5}$). We set the MAF threshold relatively high for this test because we wanted to discriminate between closely related populations with fewer samples in each. We also found an overabundance of European- or African-specific alleles (MAF ≥0.05, population threshold ≥0.25, hypergeometric test, Bonferroni-corrected $p \leq 1.42 \times 10^{-5}$). We found a similar overabundance of sex-specific alleles (MAF ≥0.05, sex threshold ≥0.40, hypergeometric test, Bonferroni-corrected $p \leq 3.01 \times 10^{-4}$); allelic variation for autosomal chromosomes is unbiased, and thus sex thresholds are closer to 0.5.

**Quantitative trait loci analysis in GEUVADIS**. Expression quantitative trait loci (eQTLs) are genetic variants that regulate RNA expression[47]; eQTL studies have been valuable for interpreting the functional significance of phenotype-associated variants from genome-wide association studies[48, 49]. Expression QTL analyses capture differences in gene expression relative to genotypes across a population, but transcript ratio QTLs (trQTLs) associate transcript ratios with genotype, deconvolving regulation of expression of each of the gene transcripts[29]. We validated the RNA isoforms that BIISQ identified in GEUVADIS by finding both cis-eQTLs and cis-trQTLs using GEUVADIS genotype data, by

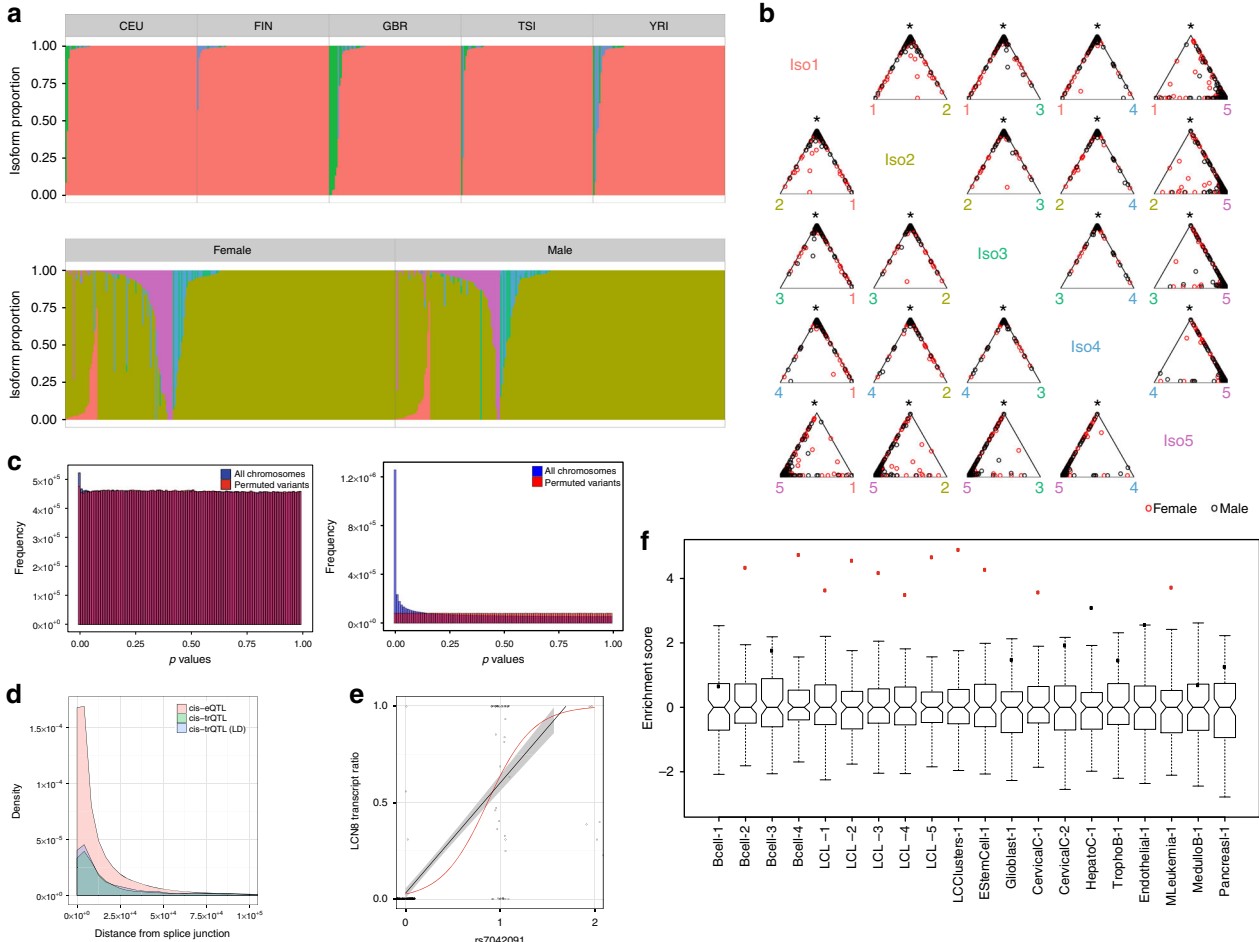

**Fig. 5** Results for isoform quantification in the GEUVADIS data. **a** The isoform quantification distribution where color denotes a unique isoform and each vertical bar is a single sample for genes *PTPRN2* (top) and *LGALS9B* (bottom). **b** Simplex plots for gene *LGALS9B* factored by sex. Each point (red for female, black for male) represents a sample's isoform composition for the two isoforms denoted on the bottom axis and the remaining isoforms at the top intersection point. **c** Matrix eQTL *p* value distribution for (left) cis-trQTLs and (right) cis-eQTLs. **d** The density of cis-trQTLs, LD pruned cis-trQTLs, and cis-eQTLs distances to the nearest canonical splice junctions in GENCODE. **e** *LCN8* contained the most significant cis-trQTL ($p \leq 2.2 \times 10^{-16}$). Linear and logistic regressions are shown in black and red. **f** Enrichment of cis-trQTLs variants in cis-regulatory annotations across cell types. Box plots show the distribution of a matched null set with Tukey whiskers (median ± 1.5 times interquartile range) and red points denoting significant enrichment (VSE test, Bonferroni-corrected $p \leq 0.01$)

separating CEU and YRI populations (following the GEUVADIS paper[29, 50], and also jointly across both populations[29, 50] (see Methods). In the CEU/YRI separated analysis, we identified 692 (458 in CEU, 250 in YRI) unique cis-trQTLs at 5% FDR compared to 639 (620 in CEU, 83 in YRI) in the GEUVADIS study[29]. Finding three times as many cis-trQTLs in YRI may be indicative of BIISQ's increased power to find lowly expressed isoforms or a relative deficiency of YRI transcripts compared to CEU in reference databases. Only 27 of the 639 genes (4.2%) identified as cis-trQTLs in GEUVADIS were also annotated as cis-trQTLs in our analysis. In total, 492 of the 692 cis-trQTLs (71.1%) in our results also had significant evidence as cis-eQTLs; in contrast, 292 of the 639 cis-trQTLs (45.7%) in the GEUVADIS study were also cis-eQTLs. This suggests that some loci may act simultaneously as a cis-eQTL and cis-trQTL, or that the regulation of transcript ratios can obfuscate gene-level quantification.

In the joint mapping of CEU and YRI data, we identified a total of 766 cis-trQTLs and 11,687 cis-eQTLs (5% FDR; Fig. 5c and Supplementary Fig. 19). We found that the cis-trQTLs, and cis-eQTLs to a lesser degree, showed spatial clustering near splice junctions (Fig. 5d). A total of 511 genes with a cis-trQTL also had a cis-eQTL (66.7%) and 264 cis-trQTLs were also cis-eQTLs

(33.33%). These results suggest that cis-trQTL signals may be masked when restricting analysis to gene-level quantification.

We computed genes with cis-trQTLs exclusively inferred by BIISQ compared to previous work[29] and evaluated their association with alternative splicing and disease phenotypes. The most significant cis-trQTL (rs7042091) was identified for gene *LCN8* by BIISQ (5% FDR; Fig. 5e); neither the gene nor the trQTL were identified in the GEUVADIS study (5% FDR)[29]. We found evidence for a cis-eQTL at this SNP-gene pair in the GTEx study across a number of tissues (whole blood, lung, muscle, spleen, and others $p \leq 2.2 \times 10^{-16}$)[49]. We also identified the *SNCA* gene among the 721 genes with a significant cis-trQTL; mutations in *SNCA*, which is typically expressed in neurons but also in LCLs, have been associated with Parkinson's disease[51]. It has been shown that alternatively spliced transcripts can cause proteins to misfold[52] and the misfolding of *SNCA*'s protein has been suggested as a therapeutic target to treat Parkinson's disease[53]. These results suggest that the alternatively spliced transcripts of *SNCA* might be interesting targets for future research and demonstrate the unique utility of BIISQ.

To further characterize the functional relationships among these cis-trQTLs, we performed variant set enrichment (VSE)

analysis for regions associated with variable intron splicing events identified by LeafCutter, which identifies regions that associate spatially with splicing QTLs[8] and cis-regulatory elements (CREs) from ENCODE in a diverse set of cell types[54–56]. VSE is a statistical test that computes the significance of enrichment or depletion of an associated variant set (here cis-trQTLs) with respect to a genomic annotation. Here, we computed enrichment of cis-trQTLs in DNase I hypersensitive sites (DHSs), H3K4me3 sites, and H3K27ac sites (Supplementary Table 3)[57]. Histone modifications are linked to the regulation of alternative splicing[58] and are associated with isoform diversity in normal and cancer cells[59]. We found that cis-trQTLs were significantly enriched in B cells when considering only DHSs (Fig. 5f: *Bcell-2,4*; VSE, Bonferroni-corrected $p \leq 2.07 \times 10^{-4}$ and $p \leq 6.39 \times 10^{-6}$) but not when jointly considering DHSs and H3K4me3 marks, or DHSs and H3K27ac marks (Fig. 5f: *Bcell-1,3*). Furthermore, cis-trQTLs were enriched in an annotation track that combines DHSs, FAIRE, and ChIP to identify regions associated with chromatin accessibility and regulatory activity in LCL samples (Fig. 5f: *LCL-1-5*; VSE, Bonferroni-corrected $p \leq 3.60 \times 10^{-3}$, $2.94 \times 10^{-5}$, $2.19 \times 10^{-4}$, $4.35 \times 10^{-3}$, $2.16 \times 10^{-5}$, respectively) and clusters of alternatively excised introns identified by LeafCutter (Fig. 5f LCClusters-1, VSE, Bonferroni-corrected $p \leq 1.57 \times 10^{-5}$). We only find significant enrichment in three of the ten control cell types, including embryonic stem cells (VSE, Bonferroni-corrected $p \leq 2.36 \times 10^{-4}$), cervical carcinoma cells (VSE, Bonferroni-corrected $p \leq 3.38 \times 10^{-3}$), and mesoderm leukemia cells (VSE, Bonferroni-corrected $p \leq 2.87 \times 10^{-3}$), indicating that there may be significant sharing of cis-trQTL-related chromatin markers between these cell types and LCLs (Fig. 5f; samples H1hescPk-1, Helas3Ifna4hPk-1, and K562Pk-1).

To understand if our cis-trQTL target genes shared biological function, we quantified enrichment of cis-trQTLs in the Database for Annotation, Visualization and Integrated Discovery (DAVID)[60] (Supplementary Table 4). We computed functional enrichment among the targets of the cis-trQTLs, and genes with >1, >4, and >6 transcript isoforms, using all annotated human genes as the background set. We found enrichment of cis-trQTL gene targets in a single KEGG pathway, olfactory transduction (BH adjusted $p \leq 2.5 \times 10^{-3}$; Supplementary Table 5). This pathway shows substantial transcript diversity: more than two thirds of olfactory receptors have been estimated to be alternatively spliced[61]. The most significant enrichment for the SwissProt and UniProt seq-feature annotations were alternative splicing and splice variant, respectively (BH adjusted $p \leq 2.2 \times 10^{-16}$ for both; Supplementary Tables 6–8). The most significant enrichment from InterPro was protein kinases (BH adjusted $p \leq 2.2 \times 10^{-16}$), which exhibit high proteomic and functional diversity as the result of alternative splicing[62]. These database enrichment results demonstrate that cis-trQTLs and spliced gene sets identified by BIISQ are enriched for alternative splicing functions and pathways.

## Discussion

We presented a statistical model, BIISQ, for quantifying RNA isoforms in short-read RNA-seq samples, which shares strength across samples to estimate isoforms—especially those at low abundance—without reference isoform compositions. We used a stochastic variational inference method to fit BIISQ to data that allows our approach to scale to transcriptome-wide study data; further, BIISQ scaled efficiently as the coverage or length of a gene increased in simulated paired-end Iso-Seq data. We demonstrated that our method improves substantially over four state-of-the-art methods in precision of isoforms on two different types of simulated data, with significant improvement for low abundance

transcripts. BIISQ also achieves relatively high recall while retaining high precision. We applied BIISQ to the GEUVADIS RNA-seq data and identified known and novel isoforms that we validated, in part, by identifying cis-trQTLs. The cis-trQTLs cluster near known splice junctions and are significantly enriched in cis-regulatory elements associated with chromatin accessibility, alternatively excised intron clusters, and histone modifications, which are all associated with splicing regulation and isoform diversity[58, 59].

BIISQ has several advantages over existing representations of RNA isoforms: (1) sample-specific isoforms are drawn from a collection of global isoforms, which leads to higher power to discover low frequency isoforms by sharing strength across samples; (2) a Bayesian hierarchical approach enables the principled incorporation of high-quality prior information such as variance in the number of known global isoforms or observed variation in the exon composition of isoforms; and (3) a non-parametric approach allows us to flexibly combine computationally tractable posterior inference with model selection, allowing the number of isoforms to grow with more samples. BIISQ also enables the interpretation of model parameters as specific quantities in RNA-seq analyses, for example, the probability of assignment of a read term to a specific isoform. BIISQ is guaranteed to converge to a local maximum, but the results on BEERS and Iso-Seq data demonstrate that the quality of isoform reconstruction is improved from taking the best maximum a posteriori solution from multiple random restarts.

Our results on GEUVADIS data show that BIISQ captures biologically interesting trends. This suggests that the partial transcript catalog identified by BIISQ can be biologically meaningful and considered for downstream analyses. The flexible and robust model for isoform identification and quantification from short-read RNA-seq data in BIISQ enables a more precise estimate of transcript isoform levels than is currently available, and opens the door to a better characterization of the cellular regulation and role of transcript isoforms in complex systems.

## Methods

**The BIISQ model.** In the BIISQ model, we consider each gene independently and a gene is defined as an ordered list of contiguous transcribed exons or retained introns; we will refer to both DNA sequence types as exons for simplicity. A gene's exons are ordered from the 5′ to 3′ end of the gene, and gene transcripts are represented by an ordered list of integers denoting the exons included in that transcript. A gene includes $\iota = 1: E$ exons, and the set of global and sample-specific isoforms are indexed by $k = 1: K$ and $\ell = 1: L$, respectively. We represent the composition of an isoform as a binary vector, where 1 signifies an exon is included, and a 0 encodes a spliced exon. Read terms are tuples defined by the terminal base pair (bp) positions of a mapped read, and the set of covered exons. For example, consider a 100 bp read that starts 40 bp into *BRCA2*, covers exons 1 and 2, and skips the first intron. The boundaries for the first two exons of *BRCA2* are [1,67] and [2617,2865], so the corresponding read term tuple is (40,2690,{1,2}).

Observations are encoded by the matrix $\mathbf{X} = (x_{vj}) \in \mathbb{R}^{V \times m}$ where $V$ is the total number of read terms observed across all $m$ samples and $x_{vj}$ denotes the number of times read term $v$ was observed in sample $j$; let $x_{\cdot j} = (x_{1j}, \ldots, x_{vj}, \ldots, x_{Vj})$ denote the vector of read term counts observed in sample $j$. The BIISQ model assumes $x_{\cdot j}$ is generated from a multinomial distribution with probability vector determined by isoform $k$, which follows a Dirichlet distribution $\boldsymbol{\beta}_k \in \mathbb{R}^V$.

$$\boldsymbol{\beta}_k \sim \text{Dir}(b_{k1}\eta_1, \ldots, b_{kV}\eta_V). \tag{1}$$

where $b_{kv} \sim \text{Bernoulli}(\pi_\iota)$ controls whether read term $v$ is expressed in isoform $k$, $\eta_1, \ldots, \eta_V$ are hyperparameters, and $\pi_\iota \sim \beta(r, s)$ if read term $v$ starts in exon $\iota$; this forms a many-to-one mapping between read terms and exons. In other words, all read terms that start in exon $\iota$ share the prior $\pi_\iota \sim \beta(r, s)$. The hyperparameters $r$ and $s$ may be tuned to encourage isoform compositions with fewer exons or, equivalently, induce sparsity over read term usage in the vector $(b_{kv})^{v=1,\ldots,V}$.

The distribution of global isoforms follows a Dirichlet process with concentration parameter $\omega$ and a uniform base distribution $H = U_\mathbb{N}$ over the set of all isoforms, $G_0 | \omega, H \sim \text{DP}(\omega, U_\mathbb{N})$. The set of natural numbers $\mathbb{N}$ defines the set of possible isoforms through their binary encodings; e.g., the number 5 encodes the isoform with the first and third exons included in a three exon gene (101). Sample-specific isoforms are distributed according to a Dirichlet process with base

distribution $G_0$, and concentration parameter $\alpha$: $G_j|\alpha, G_0 \sim DP(\alpha, G_0)$. The sharing of the base distribution $G_0$ ensures isoforms are shared among the samples, and the clustering property of the Dirichlet process encourages new observations to join existing isoforms with the largest numbers of observations (rich-get-richer property). Sample-specific and global isoforms are related through a multinomial mapping variable $c_{j,l}$, and the latent isoform assignment for each read is drawn from a multinomial distribution

$$z_{j,i} \sim \text{Multinomial}(\psi_j)$$

where

$$\psi_j = (\psi_{j,l})_{l=1}^{\infty}$$

are the sample specific isoform proportions (Supplementary Methods). Finally, reads are drawn from a multinomial distribution with probability vector determined by the global isoform, $w_{ji} \sim \text{Multinomial}(\beta_{c_{jl'}})$, $l' = z_{ji}$. See Supplementary Methods, Supplementary Fig. 20, and Supplementary Tables 9 and 10 for details of the BIISQ model.

**Posterior inference in BIISQ.** We developed a stochastic variational inference (SVI) method to tractably and robustly estimate posterior probabilities in the BIISQ model, following earlier work on SVI for the hierarchical Dirichlet process (HDP)[27]. We modified this method for the BIISQ-specific model parameters as follows. We add constant noise using hyperparameter $\epsilon$ to $b_{kv} \sim \text{Bernoulli}(\pi_t)$ which ensures that the Dirichlet distribution $\beta_k$ is defined. Sparsity, in terms of the number of exons per isoform, may be induced by controlling the hyperparameters of the beta-Bernoulli hierarchy, which affects the probability of emitting the corresponding read term in the Dirichlet distribution (Supplementary Fig. 20). Threshold parameters in the inference algorithm are configurable and tuned through the evaluation of a single held-out simulated gene (Supplementary Methods).

To handle the expansion and contraction of the population-wide isoforms, we implemented a merge-propose-reduce step in SVI and executed this step every 30 iterations[64, 65]. For every pair of isoforms, the merge step calculates the likelihood of the data before and after merging the pair of isoforms; if the sample likelihood is greater after the merge, the merge is accepted. BIISQ proposes new isoforms by computing the union of exons in randomly sampled, poorly mapped read terms, where the likelihood of that read mapping to existing isoforms is <0.5. If at least one novel isoform is proposed, BIISQ reinitializes all variational parameters (Algorithm 1 in Supplementary Methods). Finally, the reduce step removes isoform $k$ from the local and global distributions if, for all samples, there are no reads that map to isoform $k$ with a probability >0.01.

After convergence of SVI, transcripts are quantified for each sample using RPKM

$$\text{RPKM}_t = \frac{10^9 \cdot X_t}{L_t \cdot N} \qquad (2)$$

where $X_t$ is the number of reads mapped to transcript $t$, $L_t$ is the length of transcript $t$ and $N$ is the total number of mapped reads. For BIISQ, the number of reads mapped to a transcript is calculated by the product of the total number of reads mapped to the gene and the per-sample isoform proportions (Supplementary Methods).

**BEERS-simulated data runs.** Single-end RNA-seq reads were generated by the benchmarker for evaluating the effectiveness of RNA-seq software (BEERS)[16]. The number of reads required to detect a full range of isoforms in human RNA-seq experiments has been estimated to be at least 200 million[66]; therefore, we generated a pool of $(2 \times 10^8 / 2 \times 10^4) \times N_g$ reads where $N_g$ is the number of genes for each simulation. We divided the gene models into three equally sized groups according to exon count, producing groups of genes with 3–6, 7–12, and 13–182 exons. Gene models were drawn from two reference annotation data sets, the RefSeq database and a database collection composed of ten annotation tracks including UCSC and Ensembl databases (Supplementary Methods). We varied the number of novel transcripts in {2,4,6,12,16}[67], the minimum gene coverage in {1, 5, 15, 50, 100}, and the number of samples in {100, 250}. We also included 30 simulated genes for read lengths in {200, 400}. For each gene model, we sampled reads according to their parameter configuration from the aligned RNA-seq reads in the BEERS-simulated read pool. We sampled 10 genes for each parameter configuration and discarded data from BEERS-simulated genes that were under the desired coverage or did not have at least one read in each exon. This resulted in a pool of 3102 simulated genes from 532,800 RNA-seq samples. The start position of a read was sampled from a gamma distribution with a parameter that decreases linearly with the position to simulate a 5′ bias. The exonic composition of a novel splice form was generated by BEERS, and isoform proportions for each sample were sampled from a Dirichlet distribution with concentration parameter $\alpha = 1$.

**Short-read simulations from PacBio Iso-Seq long-read data.** We downloaded the full-length non-chimeric human transcriptome liver, heart, and brain data from the Iso-Seq protocol, which included unaligned sequence reads and general feature

format (GFF) reference files for each tissue[28]. The gene identifiers provided in the reference files were created independently for each tissue, so we constructed a reference set of genes and their transcripts across tissues as follows. For each gene, we created a standard set of exons by parsing its transcripts and collapsing overlapping exons in the GFF files. We then mapped genes across the three tissues based on a base pair overlap of 95% and discarded non-unique mappings. For each gene, we then mapped transcripts across tissues based on a 95% overlap (see Supplementary Methods). This process was conservative by design, leading to a confident baseline of cross-tissue isoforms. We found seven genes having at least two transcripts isoforms shared across two tissues (see Supplementary Table 2): *BLOC1S6*, *ZFAND6*, *CYTH1*, *APP*, *C1orf43*, *SPARCL1*, and *RNF14*. None of these genes were expressed in liver and thus the liver data were discarded.

We mapped the Iso-Seq reads to the gene sequences of the identified transcripts from human genome version hg19 (Supplementary Table 2) using the GMAP and STARlong algorithms[45]. We built an Iso-Seq short-read simulator (ISSRS) that simulates short reads from longer Iso-Seq reads. The inputs to ISSRS are sequencing parameters, a gene reference file with exon boundaries, and an aligned sequence read file. The outputs of ISSRS are aligned sequences in SAM format that contain short sequence reads but retain the read mapping biases present in the Iso-Seq data by copying the sequence position from the Iso-Seq reads. In brief, the simulator works as follows: (1) compute Iso-Seq reads that map to the exons of a known transcript; (2) for each read, determine the amount of sampling based on input coverage; (3) sample reads by attempting to add insert sizes distributed normally with mean 10 bp and 40 bp standard deviation; (4) output sampled reads while preserving sequence and quality scores from the aligned Iso-Seq transcripts in SAM and BIISQ format (see Supplementary Methods). For step (1), STARlong mappings yielded fewer false positives than GMAP, but GMAP produced many more usable alignments (Supplementary Figs. 9 and 10). For the seven transcripts identified across tissues, we simulated reads with lengths 50, 100, and 200 bp and approximate coverage values of the input Iso-Seq transcripts of 0.25, 0.5, and 1 for 50 samples from brain and heart tissues.

**GEUVADIS RNA-seq data preparation.** RNA-seq reads from EBV-transformed LCLs were downloaded from the Genetic European Variation in Health and Disease (GEUVADIS) project[29]. BIISQ requires read terms—mapped RNA-seq read start positions, end positions, and exons covered tuples—and a model of transcription for each gene indicating contiguous transcribed subsequences including exons or retained introns. To build the transcription model, we first extracted the protein-coding representative (as defined by ENCODE annotation "basic") transcripts from the comprehensive gene annotations in GENCODE release 19 for human genome assembly version GRCh37.p13. We then built a set of representative exons for each protein-coding gene. Most genes had a single transcript annotated as basic; for the remaining genes, we kept the transcript with the largest number of exons.

To build the read terms, we mapped the raw RNA-seq reads with STAR 2-pass to the human genome version 19. We removed unmapped reads or non-primary reads that failed quality checks or were marked as duplicates. For each mapped read, we computed the set of overlapping transcript exons, producing an intersection file. The full catalog of read terms was built from a first pass through the intersection files of each sample; we then constructed read term expression files for each sample from a second pass with the read term catalog. A final step reduces the number of read terms by collapsing terms with a similar start position and exon content to an approximate target number of read terms of 2500.

**Cis-QTL mapping.** We used Matrix eQTL[68] to perform association mapping for local eQTLs and transcript ratio QTLs (cis-trQTLs), where the ratio of expression levels for each isoform to all isoforms in a gene—or the transcript ratio—replaces the RPKM values for each gene[69]. Logistic regression (Fig. 5e) was computed using a generalized linear quasibinomial model. For the joint-processed results, we define the cis region of a gene as the genetic variants falling within 100 kb of a gene's transcription start or end site. Sex, population, the first three genotype principal components, and 15 PEER factors estimated from the isoform ratio matrix were included as covariates using a standard processing pipeline for RNA-seq data to control for population structure and latent confounders[70–72] (Supplementary Fig. 21 and Supplementary Table 11). The LD-based SNP pruned cis-trQTL set (Fig. 5d) was computed using the SNPRelate package with an LD threshold of 0.2[70]. The expression of cis-eQTLs were also quantile normalized, and we removed genes with a single transcript or fewer than three exons in the computation of cis-trQTLs. We generated the null distribution of $p$-values by permuting genotype labels while keeping isoform ratio labels constant (Supplementary Methods). To achieve well calibrated null hypothesis $p$-values and filter transcripts containing false splice junctions, transcripts with ratios of 0 or 1 in all samples and transcripts expressed in less than 10% of the samples were discarded.

For the comparisons to the GEUVADIS study, we used Matrix eQTL for association mapping with a cis region of 1 Mb, 10 PEER factors, quantile normalization, 5% FDR, and 3 genotype PCs for CEU and 2 genotype PCs for YRI. In order to accommodate partially constructed transcripts, we required only 10% of samples to have >0 expression.

**GEUVADIS functional assessment**. The Database for Annotation, Visualization and Integrated Discovery (DAVID v6.8, May 2016) analysis included functional enrichment for nine databases (Supplementary Table 4) and used the default whole genome set of genes[60]. We compiled sets of high confidence isoforms for each gene by filtering out transcript isoforms not present in ≥10% of the samples. To reduce the affects of linkage disequilibrium (LD) on the variant set enrichment analysis, we filtered genetic variants for YRI, CEU, FIN, GBR, and TSI populations such that the MAF >0.001, pairwise $r^2 < 0.8$, and genotyping rate >0.8 using the rAggr interface to Haploview on the 1000 Genomes Project phase 3 data[73]. Variant set enrichment analysis was run on the LD blocks for ENCODE Encyclopedia 3 annotations: DNase I hypersensitive sites, H3K4me3, H3K27ac, annotations generated from a chromatin state segmentation computational tool sourcing from the Broad Histone UCSC track for nine factors and nine cell types, DNase I/FAIRE/ChIP synthesis annotations from ENCODE and OpenChrom[57], and LeafCutter clusters (Supplementary Table 3). Sample Gm12878HMM was removed from enrichment analysis due to a non-normal null distribution (KS test, $p \leq 0.01$), which is required by VSE (Supplementary Fig. 22). We included annotations from LCLs as well as several other cell types: glioblastoma, cervical carcinoma, hepatocellular carcinoma, trophoblast, and embryonic stem cells.

**Population- and sex-specific splicing**. Population- and sex-specific transcript ratios were evaluated based on a likelihood ratio test. The alternative hypothesis modeled sample transcript ratios as draws from population- and sex-specific Dirichlet distributions while the null hypothesis assumed a shared Dirichlet distribution. We computed the maximum likelihood estimates for parameters of the shared, population-specific, and sex-specific Dirichlet distributions. Genes were selected for the isoform proportion plots (Fig. 5a) based on the likelihood ratio test

$$2\log \frac{\mathcal{L}(\hat{\theta}_{CEU}|x_t^{CEU}),\mathcal{L}(\hat{\theta}_{FIN}|x_t^{FIN}),\mathcal{L}(\hat{\theta}_{GBR}|x_t^{GBR}),\mathcal{L}(\hat{\theta}_{TSI}|x_t^{TSI}),\mathcal{L}(\hat{\theta}_{YRI}|x_t^{YRI})}{\mathcal{L}(\hat{\theta}_{ALL}|x_t^{ALL})} \sim \chi^2$$

where $\hat{\theta}_a$ are the maximum likelihood estimates for the parameters of the Dirichlet distribution for population $a$, $x_t$ are the sample transcript ratios and populations are denoted as superscripts. Sex-specific likelihood ratio tests were calculated analogously.

The EUR versus AFR, population-, and sex-specific variant enrichment analyses were computed from the 1000 Genomes Project phase three main release data (human genome version hg19). We describe the processing for population-specific variants (sex-specific and EUR versus AFR variants follow analogously). First, to control for linkage disequilibrium (LD), we masked variants in pairwise LD >0.90 using PLINK v1.9 (indep-pairwise 100000 1000 0.9). We then extracted two orthogonal sets of variants in protein-coding gene regions: population-specific genes (selected set) and non-population-specific genes (background set). For a specific variant, let the count of an allele $a$ in population $p$ be denoted $|a|^p$, the set of all alleles be $A$ and the set of all populations $P$. Then, a variant is population-specific if

$$\exists a, p \left| \frac{|a|^p}{\sum_{q \in P} |a|^q} > t \right. \tag{3}$$

for some population threshold $t$. Then, for each variant we count the number of population-specific alleles greater than a minor allele frequency threshold for our selected set and background set and test for an abundance of selected population-specific alleles with a hypergeometric test.

**Evaluation criteria**. An isoform transcript is defined by the set of exons that are expressed from a known gene reference. An evaluation criterion that requires the true and inferred exon sets to be identical is often conservative due to variable read coverage of exons. Therefore, isoform reconstruction was evaluated by considering both perfect and imperfect matchings to determine precision and recall (Supplementary Fig. 1). For exact matches, precision and recall were calculated based on exact full-length isoform matches between true (simulated) and estimated isoforms: let true positives, false positives, and false negatives be denoted TP, FP, and FN, respectively. Then,

$$\text{precision} = \frac{TP}{TP + FP} \quad \text{recall} = \frac{TP}{TP + FN} \tag{4}$$

For inexact matches, partial precision and recall were calculated by defining a matching $M$, or a set of pairs of inferred-true isoforms, that is of maximum cardinality and minimum weight (i.e., distance between isoform composition of a pair) between each computed transcript and the true transcripts as follows. Let $K_C$ and $K_T$ be the set of estimated and true isoforms, respectively, which are Boolean vectors of length $E$ exons {1,2, …, $E$}. A 1 at position $\iota$ signifies that exon $\iota$ is contained within that isoform, and $k[\iota]$ indexes the position of the Boolean vector $k$. We define the distance between an inferred and true isoform $d_{k,l}$ for all $k \in K_C$ and $l \in K_T$ to be the Hamming distance.

The Hamming distance counts the number of mismatched exons between the estimated and true isoforms. The maximum cardinality minimum weight $M$ is then the solution to the optimization problem

$$\min M = \sum_{k \in 1:K_C} \sum_{\ell \in 1:K_T} x_{k\ell} d_{k\ell} \tag{5}$$

$$\text{s.t.} \sum_{k \in 1:K_C} x_{k\ell} = 1 \quad \forall \ell \in 1:K_T \tag{6}$$

$$\sum_{\ell \in 1:K_T} x_{k\ell} = 1 \quad \forall k \in 1:K_C \tag{7}$$

$$x_{k\ell} \in \{0, 1\} \quad \forall k \in 1:K_C, \forall \ell \in 1:K_T \tag{8}$$

If the total number of isoforms is $I$, finding a maximum cardinality minimum weight matching can be solved in $O(I^3)$ time[74]. If $d_{k\ell}$ is the distance between inferred isoform $k \in K_C$ and true isoform $\ell \in K_T$ for matching $M$, then $d_{k,\ell} = 0$ $(d_{k,\ell}>0)$ implies $k$ is a true (false) positive; if $d_{k,\ell} \leq p|E|$ $(d_{k,\ell}>p|E|)$ then $k$ is a $p$-partial true (false) positive ($p$-TP and $p$-FP). Any true isoform not matched by a $p$-partial true positive is a $p$-partial false negative ($p$-FN). Using these definitions of $p$-TP, $p$-FP, and $p$-FN, we can compute $p$-precision and $p$-recall as in Eq. (4).

**Code availability**. The source code and software implementing the BIISQ model and inference methods can be downloaded from: https://github.com/bee-hive/BIISQ.

**Data availability**. The GEUVADIS RNA-sequencing data are available for public use in EBI ArrayExpress (accessions E-GEUV-1, E-GEUV-2, and E-GEUV-3)[29].

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

## Acknowledgements

D.A. and B.E.E. were funded by NIH R01 MH101822 and NIH R01 HL133218. A.A.P. was funded by the Jane Coffin Childs Memorial Postdoctoral Fellowship. B.E.E. was additionally funded by NIH R00 HG006265, NIH U01 HG007900, and a Sloan Faculty Fellowship. The authors gratefully acknowledge the GEUVADIS study and 1000 Genomes Project.

## Author contributions

D.A., L.-F.C., B.D., F.M., A.A.P., and B.E.E. contributed to developing the isoform reconstruction and quantification model. D.A., L.-F.C., and F.M. designed and developed the code. D.A., L.-F.C., B.D., and F.M. designed and implemented the computational

analysis pipeline. D.A. and B.E.E. designed and performed the computational analyses. D. A., A.A.P., and B.E.E. wrote the manuscript and all authors edited and approved the final manuscript.

## Additional information

**Competing interests:** The authors declare no competing interests.

