## [Peer Review File · Nature Communications]

Reviewers' comments:

Reviewer #1 (Remarks to the Author):

Discovery and quantification of isoform expression is a known to be a challenging problem. Although a large amount of RNA-seq data have been generated to date, most studies ignored isoforms. This is partly due to the lack of reliable methods for the analysis of isoforms. BIISQ offers a principled approach to discover isoforms and to quantify isoform expression levels. The main idea of BIISQ is to borrow information across samples, and this allows them to directly estimate isoform composition from the data. The authors have conducted comprehensive evaluations by both simulations as well as analysis of the GEUVADIS RNA-seq dataset. Analysis of the GEUVADIS data showed that BIISQ can facilitate the analysis of isoforms and downstream splicing QTL analysis. Below I list my major concerns of the paper.

1. Borrowing information across samples for isoform analysis is not entirely novel. Several other methods have employed similar idea:

<https://bmcbgenomics.biomedcentral.com/articles/10.1186/1471-2164-16-S2-S15>

<https://www.ncbi.nlm.nih.gov/pubmed/24307704>

It is worth including these papers in the citation.

2. Method: the notation is very confusing, which makes it difficult to understand the method.

1) Page 28, How is "read term counts" calculated? Is V the total number of read terms?

2) Page 28, equation (1) and (2) are very difficult to understand. Please clearly explain the notation in equation (1). Also, I don't see how equation (2) maps read terms to exons. This could be due to my confusion on how the read term count is calculated in the first place. Equations (1) and (2) are the critical part of the method; however, with the way it is written, I cannot understand what is done exactly. I suggest the authors to simplify the notation, make the message clear, and provide explanations for the key part of the equations.

3. Simulations:

1) The simulations are based on RefSeq annotations, but it is known that RefSeq is conservative. For the evaluations of isoform discovery, a better annotation to use is Gencode, which provides a much larger set of isoforms.

2) BEERS simulated data runs: why did you only pick $35 \times 3 = 105$ genes in the evaluation? To better understand the performance of BIISQ, it is necessary to evaluate its performance on a much larger set of genes with increased complexity.

3) Page 30, It is mentioned that "... a varying number of novel transcripts in $\{2,3,4,6\}$ ", is this the total number of transcripts of the gene? If so, this is too small.

4) It is surprising that in Figure 3, the correlation between estimated and true RPKM is so low. Most of published methods have reported much higher correlation, typically >0.8 . However, the correlation shown in Figure 3A is only around 0.5. This large discrepancy with other published papers is concerning. Further investigation is needed.

4. Evaluation criteria: for inexact matches of the estimated and true isoforms, the calculated Hamming distance counts the number of mismatched exons between the estimated and true isoforms. Simply counting an exon as yes or no may not be enough because in the extreme situation, it could be that only 1 base of an exon in the estimated isoform matches with the corresponding exon in the true isoform. If this is a large exon, apparently, 1 base of matching is not much better than random noise. A more appropriate measure would be the % of bases of the exon that match between the estimated and true isoforms. You can evaluate using different % levels, e.g., 25%, 50%, 75% etc.

5. The paper is too long. I suggest the authors to shorten the paper, move less critical information to Supplement, and make it easier for readers to understand.

Reviewer #2 (Remarks to the Author):

General Comments:

This paper proposes a Bayesian nonparametric approach for the analysis of RNA-seq data. Its contribution is in the adaptation of the hierarchical Dirichlet process model and associated variational inference, which are established machine learning methods, to the new application context of isoform detection and quantification from RNA-seq data. The method assumes that the gene loci are known but does not require annotation of gene isoforms. The method is designed specifically for joint analysis of RNA-seq data from multiple samples, which is a novel aspect of their methodology. The method was evaluated by simulation from RefSeq gene models and also simulation from isoforms obtained from PacBio long reads. Compared to current methods (Cufflinks, CEM and SLIDE), the proposed method is shown to have improved precision and recall. As expected, the improvement is particularly large for low frequency transcripts. Finally, the method is applied to analysis the RNA-seq data for the 1000 Genomes samples. This exercise demonstrated that the method is indeed feasible for large scale real data sets. However the significance of the biological findings seems modest.

Overall, this is a solid contribution to an important and current problem. It should be of interest to Nature-Comm readers.

Specific questions and suggestions:

Please provide more discussion on the reason for the improvement shown in the simulation. Is it due to the power of Bayesian nonparametric modeling and inference methodology, or is it simply because low-frequency transcripts are detected more effectively by pooling information across sample? If it is the latter reason, can we modify Cufflink, CEM or SLIDE in some simple ways to borrow information across samples? For example, and isoform with insufficient read support in one sample may ne retained and confirmed by reads from additional samples. These methods are much faster BIISQ, so if they can be adapted to achieve effective cross sample analysis, there may be substantial saving in computation.

Reference 19 and 38 are duplicated. Should one of them be a different paper?

To help users understand the computation requirement, please provide detailed information on the computational resource used for the 1000 Genomes samples, and the actual memory and CPU usage.

Reviewer: Wing Hung Wong, Stanford University

Response to reviewers

Thank you for these thoughtful and thorough reviews for this manuscript. We have addressed each point raised by the reviewers; *reviewers' comments are in italics* and our responses are in plain text. Significant text changes and additions to the manuscript were marked with **blue text** and large text removals were marked with ~~red-strike-through-text~~. Smaller stylistic or grammar changes were not marked with blue text.

Reviewer #1

Discovery and quantification of isoform expression is a known to be a challenging problem. Although a large amount of RNA-seq data have been generated to date, most studies ignored isoforms. This is partly due to the lack of reliable methods for the analysis of isoforms. BIISQ offers a principled approach to discover isoforms and to quantify isoform expression levels. The main idea of BIISQ is to borrow information across samples, and this allows them to directly estimate isoform composition from the data. The authors have conducted comprehensive evaluations by both simulations as well as analysis of the GEUVADIS RNA-seq dataset. Analysis of the GEUVADIS data showed that BIISQ can facilitate the analysis of isoforms and downstream splicing QTL analysis. Below I list my major concerns of the paper.

We thank the reviewer for a thorough description of our manuscript's contributions.

1. Borrowing information across samples for isoform analysis is not entirely novel. Several other methods have employed similar idea: <https://bmcbgenomics.biomedcentral.com/articles/10.1186/1471-2164-16-S2-S15> <https://www.ncbi.nlm.nih.gov/pubmed/24307704> It is worth including these papers in the citation.

Thank you for these references. We were not aware of them and we have incorporated them and a reference to another multiple sample isoform reconstruction method, MITIE, into the manuscript in the Introduction. Furthermore, we added the ISP method to the comparison for isoform reconstruction in the BEERS simulated data. A paragraph describing this method was added in the *Related isoform quantification methods* section of the results and reconstruction results were added to Figure 2 and multiple figures in the appendix.

2. Method: the notation is very confusing, which makes it difficult to understand the method.

We thank the reader for highlighting inconsistencies in terminology and not formally defining certain variables. These issues are now fixed in the revised manuscript.

1) Page 28, How is "read term counts" calculated? Is V the total number of read terms?

Thank you for pointing this out. We moved the definition of V from the main text to the supplement

and neglected to replace it. We have returned the definition of V to the main text and have added an exposition of “read terms” and “read term counts” (first two paragraphs of the Online Methods section). Additionally, we added a section in the supplement (Supplementary Section *Details on read term computation*) that describes how the set of read terms were calculated.

2) Page 28, equation (1) and (2) are very difficult to understand. Please clearly explain the notation in equation (1). Also, I don't see how equation (2) maps read terms to exons. This could be due to my confusion on how the read term count is calculated in the first place. Equations (1) and (2) are the critical part of the method; however, with the way it is written, I cannot understand what is done exactly. I suggest the authors to simplify the notation, make the message clear, and provide explanations for the key part of the equations.

Yes, Equations (1) and (2) were difficult to understand and are integral to the understanding of our method; we apologize for not giving additional details. We have augmented our exposition of β_k , and the vector $b_{k,v}$ which now gives additional details and intuition for β_k and $b_{k,v}$. As part of our re-written description, we simplified the definition of $b_{k,v}$ by completely removing the mapping function g . The description of ϵ was moved and clarified in the *Posterior inference in BIISQ* section of the Online Methods. .

3. Simulations: 1) The simulations are based on RefSeq annotations, but it is known that RefSeq is conservative. For the evaluations of isoform discovery, a better annotation to use is Gencode, which provides a much larger set of isoforms.

We likely did not explain the BEERS simulator well enough in the manuscript; we have updated the description carefully including additional details in the Supplementary Methods section *BEERS simulations*. The BEERS simulator uses reference annotations of gene models that indicate which exons are transcribed, to construct novel isoforms. So, for any particular gene selected by BEERS, the main transcript will indeed be copied from the reference annotation, but the novel isoforms will be computationally generated.

Your point is well-taken, and it is possible that the variety of isoforms observed in more liberal annotations could be selected as reference transcripts in BEERS by adding complexity to the simulation. The BEERS simulation software includes configuration files for RefSeq and for an ensemble configuration made from ten annotation tracks: UCSC, RefSeq, RefSeq-Other, Ensembl, Vega, AceView, GenScan, GeneID, NSCAN, and SGP. For the additional runs we describe here, we use this set of annotation tracks (which draws from 538,991 total gene models and much of GENCODE via Ensembl).

We should also mention that we did consider this idea when selecting GENCODE to use for the annotations to build the gene model for the GEUVADIS results. Further, the three major computational studies we performed for model validation used distinct references: (1) the BEERS simulations are based on RefSeq (and now the Ensembl annotations), (2) the PacBio simulations are based on PacBio's internal transcript calling pipeline, and (3) GEUVADIS is based on GEN-

CODE. We believe showing promising results from a diverse set of data sources is a strength of our method.

2) *BEERS simulated data runs: why did you only pick 35x3 = 105 genes in the evaluation? To better understand the performance of BIISQ, it is necessary to evaluate its performance on a much larger set of genes with increased complexity.*

You raise the good point that we may be undersampling genes and not capturing the complexity of the human transcriptome. The modified BEERS simulations increase the number of sampled genes from 105 to 3,102, with approximately half of them taken from the RefSeq annotation and half from the Ensembl annotation described in the previous response. This resulted in 532,800 simulated individuals compared to 10,500 previously. We updated the results in *RNA-sequencing simulation: BEERS*, Figure 2, Figure 3 (top half), Supplementary Figs. 2-6 and the methods description in the *BEERS simulated data runs* section).

3) *Page 30, It is mentioned that "... a varying number of novel transcripts in 2,3,4,6", is this the total number of transcripts of the gene? Is so, this is too small.*

We increased the number of novel transcripts per gene from {2,3,4,6} to {2,4,6,12,16} according to estimates from recent work that describes the "upper plateau" for the number of transcripts per gene to be 10-12 [2]. This change represented the largest difficulty in inference for isoform discovery and quantification (Supplementary Figure 3). The updates to the manuscript reflecting this change are found in the BEERS related sections.

4) *It is surprising that in Figure 3, the correlation between estimated and true RPKM is so low. Most of published methods have reported much higher correlation, typically > 0.8. However, the correlation shown in Figure 3A is only around 0.5. This large discrepancy with other published papers is concerning. Further investigation is needed.*

While we agree that the correlation between inferred and true *gene* expression – and, in some cases, *isoform* expression in simpler data – reaches this level of correlation, this does not seem to be the case for correlation between inferred and true *isoform* expression in complex data. In general, the correlation between inferred and true *isoform* expression is principally a function of the complexity of the data. This is supported by the fact that the correlation between true and estimated RPKM is comparatively low across all methods in the BEERS simulations. However, we note that the correlation for methods with high precision and low recall tends to be higher, and, importantly, it is much higher for BIISQ and Cufflinks in the PacBio data. It is considerably more difficult to infer isoforms in the new BEERS data due to the increased number of alternative splice forms per gene and the low isoform coverage for a large portion of the genes (see the new Figure titled *Average coverage per gene density plot*. in the Supplementary Materials). Supplementary Fig. 6 shows that, if you restrict the results to genes with high coverage, the correlation is increased, and the increase is dramatic for some methods (e.g., up to 0.933 for BIISQ).

One of our primary references which compares isoform reconstruction and quantification methods was Steijger *et al.* [3]. Their evaluation of the correlation between inferred and true expression is

very much in line with our estimates in, arguably, simpler data than we simulate. The phenomenon of lower correlation in isoforms compared to genes appears to be true even when comparing expression levels across experimental platforms [1].

4. Evaluation criteria: for inexact matches of the estimated and true isoforms, the calculated Hamming distance counts the number of mismatched exons between the estimated and true isoforms. Simply counting an exon as yes or no may not be enough because in the extreme situation, it could be that only 1 base of an exon in the estimated isoform matches with the corresponding exon in the true isoform. If this is a large exon, apparently, 1 base of matching is not much better than random noise. A more appropriate measure would be the % of bases of the exon that match between the estimated and true isoforms. You can evaluate using different % levels, e.g., 25%, 50%, 75% etc.

We were apprehensive about defining a measure based on percent of exons covered because it would favor annotation-based methods like BIIISQ. But, for completeness, we defined an exon coverage measure and generated results for each of the methods applied to BEERS data. We have included this measure in the main text (Section *RNA-sequencing simulation: BEERS*), a detailed description in the Supplementary Section *Statistics for evaluation*, and results in Supplementary Table 9.

5. The paper is too long. I suggest the authors to shorten the paper, move less critical information to Supplement, and make it easier for readers to understand.

We dramatically shortened the manuscript by removing text in red or moving them to the Supplementary Materials. Briefly, this includes a description of how each method was run, and the function that maps read terms to exons and its description. Additionally, we have made many changes to aid reader comprehension that have been marked in blue text in the revised manuscript. Most notably, we

1. Added a description and results for the ISP method.
2. Rewrote the paragraphs describing the BIIISQ model in the Online Methods.
3. Rewrote the BEERS simulation description and results to reflect the new simulations.
4. Added a new metric based on exon coverage.
5. Added or changed several figures in the Supplementary Materials.
6. Described how BIIISQ is run from aligned sequence data.

Reviewer #2

This paper proposes a Bayesian nonparametric approach for the analysis of RNA-seq data. Its contribution is in the adaptation of the hierarchical Dirichlet process model and associated variational inference, which are established machine learning methods, to the new application context of isoform detection and quantification from RNA-seq data. The method assumes that the gene loci are known but does not require annotation of gene isoforms. The method is designed specifically for joint analysis of RNA-seq data from multiple samples, which is a novel aspect of their methodology. The method was evaluated by simulation from RefSeq gene models and also simulation from isoforms obtained from PacBio long reads. Compared to current methods (Cufflinks, CEM and SLIDE), the proposed method is shown to have improved precision and recall. As expected, the improvement is particularly large for low frequency transcripts. Finally, the method is applied to analysis the RNA-seq data for the 1000 Genomes samples. This exercise demonstrated that the method is indeed feasible for large scale real data sets. However the significance of the biological findings seems modest. Overall, this is a solid contribution to an important and current problem. It should be of interest to Nature-Comm readers.

We thank the reviewer for an accurate summary of our manuscript. We hope the responses below address the reviewer's concerns with the significance of the biological findings.

Please provide more discussion on the reason for the improvement shown in the simulation. Is it due to the power of Bayesian nonparametric modeling and inference methodology, or is it simply because low-frequency transcripts are detected more effectively by pooling information across sample? If it is the latter reason, can we modify Cufflink, CEM or SLIDE in some simple ways to borrow information across samples? For example, and isoform with insufficient read support in one sample may ne retained and confirmed by reads from additional samples. These methods are much faster BIISQ, so if they can be adapted to achieve effective cross sample analysis, there may be substantial saving in computation.

The improvements shown in the manuscript are likely due to a combination of several factors, and we have added this update to the manuscript. As you state, there are advantages to the Bayesian paradigm: it enables encoding the prior beliefs of the exon segments and prior probabilities of including a given exon, which are useful in scenarios where you can approximate exon usage. In this work, we set the hyperparameters by grid search of a single gene, which we now describe in Supplementary Section titled *Parameter settings for each method*. We have also expanded the Discussion to make the benefits of model parameter interpretability more clear.

For generalizing methods to borrow information across samples, Cufflinks and similar methods build an overlap graph from the aligned RNA-seq reads. This overlap graph encodes which fragments are likely adjacent in some subset of isoforms given the aligned fragments. The problem then becomes finding paths through this overlap graph. Each path is consistent with some subset of reads, and forms an isoform transcript.

Firstly, there may be computational problems associated with generalizing this approach to mul-

multiple individuals. Given that the overlap graph is acyclic, the minimum path cover problem can be reduced to a matching problem that can be solved in $O(V^2E)$ time, where V is the number of vertices in the overlap graph and E are the number of edges. This is relatively fast. However, the optimization procedure required for simultaneous modeling of multiple samples in this overlap graph is not well defined. A computationally feasible approach would have to include a sample clustering step on top of path finding to group samples with similar expression patterns.

Secondly, computational challenges would likely arise from the rapid increase of the number of possible isoforms that could arise after including the erroneous mappings and isoform transcripts that exist in all samples. Thus, computing the set of all possible combinations of paths that are consistent with the overlap graph could be prohibitive.

For completeness, we described and computed results for an algorithm that does implement multi-sample analysis: ISP [4]. Please see the BEERS results and description of methods sections (Figure 2 and BEERS Supplementary Figs. 2-4).

Reference 19 and 38 are duplicated. Should one of them be a different paper?

Thank you for catching this error. Indeed, there was a duplicate reference for IsoLasso; we have updated the manuscript.

To help users understand the computation requirement, please provide detailed information on the computational resource used for the 1000 Genomes samples, and the actual memory and CPU usage.

We have added additional computational details for the GEUVADIS runs (Section *Quantitative trait loci analysis in GEUVADIS*. and Supplementary Fig. 21) that should give readers an idea of the computational cost of running BISSQ on RNA-seq data.

In addition to the reviewer comments, we have also fixed formatting issues in the supplement and references sections.

References

- [1] Matthew Dapas, Manoj Kandpal, Yingtao Bi, and Ramana V Davuluri. Comparative evaluation of isoform-level gene expression estimation algorithms for RNA-seq and exon-array platforms. *Briefings in Bioinformatics*, 18(2):260–269, 2016.
- [2] Sarah Djebali et al. Landscape of transcription in human cells. *Nature*, 489(7414):101–108, September 2012.
- [3] Tamara Steijger et al. Assessment of transcript reconstruction methods for RNA-seq. *Nature Methods*, 10(12):1177–1184, Dec 2013.

- [4] Masruba Tasnim, Shining Ma, Ei-Wen Yang, Tao Jiang, and Wei Li. Accurate inference of isoforms from multiple sample RNA-seq data. *BMC Genomics*, 16(2):S15, Jan 2015.

Reviewers' comments:

Reviewer #1 (Remarks to the Author):

I thank the authors for taking into account most of my comments. I think the paper is now much clearer and easier to read. I however still have a major comment on data simulation and method evaluation.

From Figure 3, it appears that the simulated data have very high RPKMs. The range is from 100 to 100000, which seems unrealistic. In real data, many isoforms have low to moderate expression, even 100 RPKM is considered high. To provide a fair comparison with other methods, the authors should simulate data with realistic RPKMs. Since the authors have analyzed multiple real RNA-seq datasets, you could use those real data to guide the simulations.

Reviewer #2 (Remarks to the Author):

The authors have provided satisfactory responses to my comments. I am happy with the revised paper and think that it is ready for publication.

One minor typo: page 8, first sentence of second paragraph: "The first evaluated our model" should be "We first evaluated our model"

Response to reviewers

Thank you for the additional reviews for this manuscript. We have addressed the points raised by the reviewers below. Significant text changes and additions to the manuscript from the previous version were marked with blue text.

Reviewer #1

From Figure 3, it appears that the simulated data have very high RPKMs. The range is from 100 to 100000, which seems unrealistic. In real data, many isoforms have low to moderate expression, even 100 RPKM is considered high. To provide a fair comparison with other methods, the authors should simulate data with realistic RPKMs. Since the authors have analyzed multiple real RNA-seq datasets, you could use those real data to guide the simulations.

Thank you for pointing a potentially misleading aspect of Figure 3. We agree that only simulating transcripts at high coverage would not be a fair representation of experimental data. In fact, most transcripts in the BEERS data were simulated at low coverage (Supplemental Figure 5). Indeed, 25.7% of the genes had an average coverage across samples and isoforms of less than 1; 56.7% had coverage less than 5; 71.2% had coverage less than 10. We updated the text to reflect this coverage.

The RPKMs seem inflated because each method was run on the aligned sequence reads for each gene independently. Therefore, RPKM is inversely proportional to the number of reads in each gene rather than in the experiment. We did this for two reasons: (1) our method takes as input sequence reads that align to a single reference transcript, and, to be consistent, we also ran other methods with sequence reads from a single transcript. (2) Some methods performed better when we restricted the input data to only those reads that aligned to a single reference transcript. To more appropriately represent these RPKM values, we rescaled each RPKM by the ratio of the number of reads aligning to the gene to the average number of reads in the GEUVADIS experimental data. We also changed Pearson correlation to Spearman because non-linear relationships between inferred and true RPKM arise in methods that fail to reconstruct isoforms or misalign reads. Spearman correlations better captures a monotone trend. We added text to explain that RPKM was calculated independently for each gene and normalized post hoc, and a sentence explaining that most transcripts were simulated at low coverage, which references Supplemental Figure 5.

Reviewer #2

One minor typo: page 8, first sentence of second paragraph: "The first evaluated our model" should be "We first evaluated our model"

Thank you, we made the appropriate change.

REVIEWERS' COMMENTS:

Reviewer #1 (Remarks to the Author):

Thanks for clarification on the RPKM calculations. I don't have any additional concerns.